# Is a Drug Allergy in a Patient’s History Real? Our Experience with Diagnostic Drug Provocation Tests

**DOI:** 10.3390/medicina61030386

**Published:** 2025-02-23

**Authors:** Begum Gorgulu Akin, Betul Ozdel Ozturk, Makbule Seda Bayrak Durmaz, Ozge Ozturk Aktas, Sadan Soyyigit

**Affiliations:** 1Department of Immunology and Allergic Diseases, Ankara Bilkent City Hospital, 06800 Ankara, Turkey; drbegumgorgulu@gmail.com (B.G.A.); betulozdel84@gmail.com (B.O.O.); dr.seda_bayrak@hotmail.com (M.S.B.D.); 2Division of Allergy and Clinical Immunology, Ankara Bilkent City Hospital, School of Medicine, Ankara Yildirim Beyazit University, 06800 Ankara, Turkey; doctorozge@hotmail.com

**Keywords:** drug hypersensitivity reactions, diagnostic drug test, suspected drug

## Abstract

*Background and Objectives:* Early-type drug hypersensitivity reactions (DHRs) are observed within the first 1–6 h and most commonly manifest as urticaria and/or angioedema. Detailed anamnesis, skin prick tests (SPTs), intradermal tests (IDTs), and oral/intramuscular/intravenous drug provocation tests (DPTs) can be used to identify the drug responsible. We aimed to evaluate the demographic characteristics, responsible drugs, DHR types, and DPT results used in the diagnosis of drug allergy in patients who presented to our clinic with suspected drug allergies. *Materials and Methods:* The medical records of patients who presented with a suspicion of an early-type DHR between February 2019 and December 2024 were retrospectively evaluated through the hospital information management system. A total of 188 adults who underwent diagnostic drug testing were included. *Results:* The diagnosis of drug allergy was confirmed in 51 (27%) patients. In 137 (73%) patients, the diagnosis of drug allergy was excluded after DPTs. In 78 of the 188 patients, there was a DHR to a single suspected drug. The other 110 patients had DHR histories with multiple drugs. The rate of confirmation of a drug allergy from diagnostic tests was higher in those who described a history of multiple drug allergies. Amongst the antibiotics, beta-lactam antibiotics (*n* = 47) were the most frequently suspected drugs. The rate of positive DPTs (*n* = 4; 8%) was lower in patients with suspected beta-lactam allergies than other antibiotics (*p* = 0.002). NSAIDs (*n* = 60) were the second most common group of suspected drug allergies. With regard to IgE or COX-1-mediated mechanisms, there was no statistically significant difference in DPT positivity among these NSAIDs (*p* = 0.414). The severity of the initial early-type DHRs were grade 1 (*n* = 168; 80%), grade 2 (*n* = 14; 7%), and grade 3 (*n* = 14; 7%). If the patients had redness, itching, urticaria, angioedema, dyspnea, cyanosis, desaturation, syncope, tachycardia, or hypotension during their initial DHRs, the positive diagnostic drug test rate was statistically significantly higher. However, experiencing diarrhea, nausea, and vomiting were not found to be associated with positive diagnostic drug tests. Drug allergies were confirmed with SPTs or IDTs in all patients in whom adrenaline was used during initial reactions. *Conclusions:* Contrary to the prevailing notion that drugs (especially beta-lactams) are the predominant cause of allergic reactions, this study demonstrated that the actual prevalence of drug allergies is, in fact, low.

## 1. Introduction

The unwanted and harmful effects that occur with normal doses of medicines are called adverse drug reactions (ADRs). These reactions are divided into two main groups, type A and type B. Type A reactions, which constitute the majority of reactions and can occur in anyone who uses the drug, are predictable and dose-dependent. Adverse, toxic, and indirect effects of the drug and drug interactions make up these reactions. Type B reactions are unpredictable and dose-related or independent reactions, accounting for approximately 15–20% of ADRs [1]. Type B reactions are also called drug hypersensitivity reactions (DHRs) [2]. DHRs are divided into two subgroups: allergic [immunoglobulin E (IgE) and non-IgE mediated] and non-allergic. A significant proportion of ADRs are early-type IgE-mediated or T cell-mediated late reactions [3,4]. Early-type DHRs are observed within the first 1–6 h and most commonly manifest as urticaria and/or angioedema. Drug-induced anaphylaxis is also considered part of this group [5].

Drug allergy has become a major public health problem with the increasing use of medications. Data on the true incidence of DHRs are limited. It is estimated that approximately 3–5% of all patients admitted to hospital and 10–15% of hospitalized patients have a drug allergy [6,7,8]. One of the greatest difficulties in the detection of drug allergies is the similarity between the clinical symptoms of the underlying disease and the symptoms caused by drugs. Another problem is the difficulty in identifying the responsible drug due to the history of multiple drug use in the anamnesis of patients presenting with drug allergies [2,5,6,7].

Detailed anamnesis, skin prick tests (SPTs), intradermal tests (IDTs), and oral/intramuscular (IM)/intravenous (IV) drug provocation tests (DPTs) can be used to identify the responsible drug [2,6,9,10]. DPTs are the gold standard method for the definitive diagnosis of drug allergy. In patients presenting with drug allergy, it is very important to identify the drug causing the allergy, determine alternative drugs that can be used after banning the responsible drug, organize treatments to be taken by the patient afterwards, and give the patient a medication card.

In this study, we aimed to evaluate the demographic characteristics, responsible drugs, DHR types, and DPT results used in the diagnosis of drug allergy in patients who presented to our clinic with suspected drug allergies.

## 2. Materials and Methods

### 2.1. Study Design

The medical records of patients who presented with a suspicion of early-type DHRs and underwent alternative or diagnostic DPTs between February 2019 and December 2024 were retrospectively evaluated through the hospital information management system. A total of 188 adults who underwent diagnostic drug testing were included in this study, in accordance with the tenets of the Declaration of Helsinki, after receiving approval from the local ethics committee of Ankara Bilkent City Hospital TABED Ethics Committee (Approval number: TABED 1-24-828).

Patient medical records were analyzed and the following information was obtained: age, sex, suspected drug(s), reaction and time of occurrence, the route of administration of the suspected drug, history of atopy, concomitant allergy and other diseases, treatments administered, and diagnostic investigations. This information was evaluated retrospectively. Reactions observed up to 6 h after the last dose were considered early-type DHRs. This study excluded patients aged under 18 years and those with T cell-mediated late reactions.

### 2.2. Severity of Early-Type Drug Hypersensitivity Reactions

Early-type DHRs were classified as mild (grade 1), moderate (grade 2), or severe (grade 3) in accordance with Brown’s grading system. The signs and symptoms of DHRs were defined as cutaneous (flushing, pruritus, urticaria, angioedema), cardiovascular (chest pain, tachycardia, presyncope, syncope, and hypotension), respiratory (nasal–ocular symptoms, dyspnea, wheezing, oxygen desaturation, and throat tightness), gastrointestinal (nausea, vomiting, diarrhea, and abdominal pain), and atypical manifestations (fever/chills, back and neck pain, and numbness/weakness) [11].

### 2.3. Specific Diagnostic Evaluation of Beta-Lactam Antibiotics

The presence of ampicillin-, amoxicillin-, penicillin G-, and penicillin V-specific IgE (sIgE) was investigated using the Immulite 2000 system (Siemens Healthcare Diagnostics, Tarrytown, NY, USA) in patients with a history of reactions to beta-lactam antibiotics, and values above 0.35 kUA/L were considered positive [12]. If the sIgE was negative, a specific standardized penicillin allergenic determinant skin test (DAP^®^, Diater Laboratories, Madrid, Spain) was performed [13,14]. Skin tests were performed using penicillin major determinant [Penicillin polylysine (PPL) and minor determinant mixture (MDM) [penicilloate, penilloate]. Afterwards, SPTs and IDTs were performed using penicillin G or amoxicillin/amoxicillin clavulanate according to the suspected drug reaction in patients who underwent major and minor determinant tests. Histamine (10 mg/mL) was used as the positive control and 0.09% sterile saline was used as the negative control. If skin tests were negative and there was no history of life-threatening anaphylaxis, the testing was continued with a DPT. Beta-lactam antibiotic DPTs were performed according to the protocols in the European Network on Drug Allergy/European Academy of Allergy and Clinical Immunology (ENDA/EAACI) guidelines [15].

### 2.4. Specific Diagnostic Evaluation of Non-Steroidal Anti-Inflammatory Drugs

Acetyl salicylic acid (ASA) provocation tests were performed in patients who had early reactions to more than one suspected non-steroidal anti-inflammatory drug (NSAID) that were not chemically similar but had the common feature of a pharmacologic inhibition of cyclooxygenase-1 (COX-1). DPT was performed with a placebo on the first day of the test and with ASA on the second day. During oral ASA provocation tests, patients were observed for lower respiratory symptoms such as shortness of breath, wheezing, and chest tightness; upper respiratory symptoms such as a runny nose and nasal congestion; and general symptoms such as watery eyes, swelling around the eyes, redness of the skin, and urticaria. The test was considered positive if their forced expiratory volume (FEV1) or peak expiratory flow (PEF) fell by more than 20%, and respiratory symptoms and/or accompanying extrabronchial symptoms were very intense [16]. Patients with a positive ASA provocation test were forbidden to take NSAIDs with strong COX-1 inhibition, and then DPTs were performed on NSAIDs with a weak inhibition of COX-1 (i.e., paracetamol < 1000 mg), preferential COX-2 inhibitors (i.e., nimesulide and meloxicam) or selective COX-2 inhibitors (celecoxib) [16,17]. An IgE-mediated NSAID allergy was considered in patients with a history of reactions to a single drug or more than one drug that was chemically similar. In patients with a suspected IgE-mediated NSAID allergy, DPTs were performed according to the ENDA/EAACI guidelines [9,16]. Detailed information about these diagnostic drug testing protocols is given in the following paragraph.

### 2.5. Diagnostic Testing Protocols for Other Drugs

Patients with no contraindications for DPTs were informed about the test in detail and consent was obtained from each patient. Antihistamines were discontinued 5 days before and steroids were discontinued approximately 2 weeks before DPTs, depending on the dose and strength of the drugs. Skin tests were performed according to the protocols in the ENDA/EAACI guidelines [9,10]. Histamine (10 mg/mL) was used as the positive control and 0.09% sterile saline was used as the negative control. SPTs were performed with the maximum non-irritant concentration of the drug. A mean wheal diameter of ≥3 mm obtained with the control solution was considered a positive result. If the SPT of the suspected drug was negative, an IDT was performed in increasing doses until the maximum non-irritant concentration was reached. The IDT was regarded as positive if the initial wheal increased by at least 3 mm in diameter and was surrounded by erythema after 20 min [10]. Patients with positive skin tests were considered to be allergic to the suspicious drug and a DPT was not performed. If skin tests were negative and there was no history of life-threatening anaphylaxis, the testing was continued with a DPT. All patients were first given a placebo before the test. All DPTs were performed under hospital conditions and after emergency intervention facilities were provided.

Before the DPTs, the vital signs and PEF values of the patient were recorded and vascular access was established. In oral DPTs, one-quarter of the drug was administered first. After 60 min, if there were no symptoms, the remaining three-quarters of the drug was administered. In IV or SC DPTs, the test was started by administering one-tenth of the drug first and then the required dose was reached by increasing the dose in stages. Patients who were able to take the last dose of the drug without any problems were kept under observation for at least 2 h. In the absence of symptoms and variability in their PEF values, the test was considered negative and it was confirmed that the patient could safely take the drug [9,10].

The test was considered positive and terminated if any of the following symptoms occurred during or after the DPT: cutaneous symptoms (urticaria, angioedema), cardiovascular system symptoms (hypotension, tachycardia), respiratory system symptoms (cough, wheezing, shortness of breath, more than a 20% decrease in FEV1 compared with baseline), neurologic system symptoms (confusion, syncope), and gastrointestinal system symptoms (abdominal pain, vomiting, diarrhea). Patients were given the necessary medical treatment and kept under observation until all findings improved. A few weeks later, the patient was tested with alternative drugs and given a drug information card.

### 2.6. Statistical Analysis

In this study, power analysis was conducted utilizing G Power 3.1, and sample analysis was performed. The following parameters were used for the power analysis: effect size = 0.5; alpha error probability = 0.05; power (1−β error probability) = 0.8; and allocation ratio N2/N1= 3. The total sample size was calculated to be 170.

Data analysis was performed using the SPSS 11.5 for Windows software package (SPSS Inc., Chicago, IL, USA). Descriptive statistics for nominal data are presented as counts and percentages, and quantitative data are presented either as means ± standard deviations or medians and the minimum–maximum depending on assumptions of normality based on visual (histograms and probability graphs) and analytical methods (Kolmogorov–Smirnov and Shapiro–Wilk tests). The Chi-square or Fisher’s exact test was used to compare categorical variables as appropriate. All *p*-values below 0.05 were considered significant.

## 3. Results

### 3.1. General Findings in the Study Population

A total of 1640 patients underwent alternative or diagnostic DPTs in allergy and clinical immunology clinics between February 2019 and December 2024. A total of 188 patients underwent diagnostic drug allergy testing to confirm or exclude their drug allergy. After a diagnostic evaluation, the diagnosis of drug allergy was confirmed in 51 (27%) patients. In 137 (73%) patients, the diagnosis of drug allergy was excluded after DPTs (Figure 1). DPTs were performed, using a total of 252 drugs, in 137 patients in whom drug allergy was ruled out. Among these patients, amoxicillin clavunate was the most frequently used drug, and a total of 30 DPTs were performed. The second most frequently used drug was ASA (*n* = 26). The results are shown in detail in Figure 2.

Of the 188 patients, 146 were women. Drug allergy was confirmed in 26% *(n* = 38) of the women and 31% (*n* = 13) of the men in this study. There was no statistically significant difference between the men and women in either group (*p* = 0.327). The median age of the patients with negative diagnostic drug tests was 37 (18–70) years, and the median age of the patients with positive diagnostic drug tests was 43 (range, 21–65) years (*p* = 0.147). Housewives were the largest group of patients who underwent a diagnostic drug test. Interestingly, the rate of negative drug test results was statistically significantly higher in students (*p* = 0.032). The findings revealed no statistically significant difference between educational attainment and the outcomes of DPTs (*p* = 0.616). In addition, no statistically significant difference was observed in the results of the drug tests with regard to comorbidities, a history of allergic diseases, or atopy (*p* = 0.442, *p* = 0.491, and *p* = 0.704, respectively). Detailed results are presented in Table 1.

### 3.2. Evaluation of Suspected Drugs in Patients

In 78 of the 188 patients there was a history of reactions to a single suspected drug. The other 110 patients had DHR histories with multiple drugs. The rate of confirmation of allergy through positive DPTs was statistically significantly lower in patients with a history of reactions to a single suspected drug than in patients with histories of reactions to multiple suspected drugs (*p* = 0.046). The median number of suspected drugs in patients with positive diagnostic drug tests was 2 (range, 2–5), and the mean number of suspected drugs in patients with negative tests was 1 (range, 1–5) (*p* = 0.010). The most prevalent category of suspected drug allergy documented in patients’ medical histories was antibiotics (*n* = 61). Amongst the antibiotics, beta-lactam antibiotics (*n* = 47) were the most frequently suspected drugs. The most common suspected allergy among beta-lactam antibiotics was to amoxicillin/clavulanate (*n* = 35). There was no significant difference in DPT positivity between the different subgroups of the beta-lactam antibiotic group (*p* = 0.615) (Table 1).

In patients with a suspected drug allergy to antibiotics, a history of reactions to multiple antibiotics was the second most common (*n* = 22). The third most common was a suspected allergy to quinolone antibiotics (*n* = 5). Despite the limited number of patients with a suspected quinolone allergy, our statistical analysis revealed that the rate of positive DPTs (*n* = 4; 80%) was statistically significantly higher in patients with a suspected quinolone allergy. Conversely, the rate of positive DPTs (*n* = 4; 8%) was lower in patients with a suspected beta-lactam allergy (*p* = 0.002) (Table 1).

NSAIDs (*n* = 60) were the second most common group of suspected drug allergies. The most frequently suspected mechanism of their DHRs was COX-1-related (*n* = 50). With regard to the IgE- or COX-1-mediated mechanisms, there was no statistically significant difference in DPT positivity among these NSAIDs (*p* = 0.414). Detailed results about these patients’ suspected drug allergies are given in Table 1.

### 3.3. Evaluation of Characteristics Relevant to Initial Early-Type Drug Hypersensitivity Reactions

The severity of initial early-type DHRs were grade 1 (*n* = 168; 80%), grade 2 (*n* = 14; 7%), or grade 3 (*n* = 14; 7%) (Figure 3a). The most common symptoms reported by patients during reactions were related only to cutaneous involvement (*n* = 137; 73%). Anaphylaxis symptoms were reported to be the second most common (*n* = 16; 8.5%). The least common symptoms were related to the gastrointestinal system (*n* = 2; 1%) (Figure 3b). In our study, it was observed that the diagnosis of allergy was confirmed through diagnostic tests in 13 of 14 patients with grade 3 initial DHRs. The rate of positive drug tests was statistically significantly higher in patients with a history of grade 3 DHRs (*p* = 0.001).

The median reaction time after taking a drug was 15 (range, 2–300) minutes in patients with positive diagnostic drug tests and 60 (range, 2–600) minutes in negative tests, which was statistically significantly different (*p* = 0.001). Patients that had redness, itching, urticaria, angioedema, dyspnea, cyanosis, desaturation, syncope, tachycardia, or hypotension during their initial DHRs had a statistically significantly higher rate of positive diagnostic drug tests. However, experiencing diarrhea, nausea and vomiting were not found to be associated with positive diagnostic drug tests. The related *p*-values are given in Table 2. In addition, diagnostic drug tests were statistically significantly negative if the patients had atypical symptoms such as feeling sick (*p* = 0.030). Drug allergies were confirmed through SPTs or IDTs in all patients in whom adrenaline was used during initial early-type DHRs (*p* = 0.001) (Table 2).

### 3.4. Results of Diagnostic Drug Testing Protocols

All patients were started on a placebo before the test. Of the 188 patients, 16 had atypical symptoms with the placebo. None of the patients symptomatic to the placebo drug had a positive diagnostic drug test (*p* = 0.007) (Table 2).

After evaluations, the diagnosis of drug allergy was confirmed in 51 (27%) patients. The diagnosis of drug allergy was verified through positive skin tests in 35% (*n* = 20) of these patients. In four patients, this diagnosis was based on positive drug SPTs. Two of these patients were positive for midazolam, one for fentanyl, and one for lidocaine. IDT positivity confirmed the diagnosis of drug allergy in 16 patients. In half (*n* = 8) of these patients, their IDTs were positive for general anesthetic drugs. Among the general anesthetic drugs tested, IDT positivity was found most frequently for rocuronium. All these drug sensitivities were compatible with the drug history of the patients in their early-type DHRs. Therefore, patients with positive skin tests did not undergo DPTs. Patients with a suspected allergy to general anesthetic drugs but negative skin tests were excluded from this study because DPTs could not be performed under outpatient clinic conditions. Details of the skin test results for the patients’ suspected drugs are given in Figure 1.

DPTs were performed if the skin test results were negative for the drugs when using the skin test protocols in the guidelines. For drugs without a skin test protocol or NSAIDs for which COX-1-mediated DHRs were considered, direct drug provocation tests were performed without SPTs and IDTs. In 31 patients, DPTs were performed. ASA provocation tests were positive in six patients. Despite nimesulide (*n* = 1) and meloxicam (*n* = 2) being preferential COX-2 inhibitors, three patients displayed allergic symptoms during their DPTs due to COX-1-mediated hypersensitivity. The diagnosis of Ig E-mediated drug allergy was confirmed in six patients who described having allergic reactions to acetaminophen (*n* = 3), diclofenac (*n* = 1), ibuprofen (*n* = 1), and etodolac (*n* = 1). The presence of grade 1 DHRs was observed in these patients during their DPTs. In addition, ASA provocation tests performed to exclude COX-1-mediated mechanisms in patients with IgE-mediated NSAID DHRs were negative.

Amoxicillin-specific IgE was found to be positive in only one patient through IgE measurements for the diagnosis of a DHR to beta-lactam antibiotics. This patient was given an oral amoxicillin/clavulanate DPT as their initial DHR indicated low risk. A grade 1 DHR was seen during their DPT.

In 12 patients, skin tests were performed using DAP diagnostic sets [penicillin major determinant and MDM (penicilloate, penilloate)]. Afterwards, SPTs and IDTs were performed with penicillin G or amoxicillin/amoxicillin clavulanate according to the suspected drug reaction in patients who underwent major and minor determinant tests. Their skin tests with beta-lactams were negative and then DPTs were performed. No drug allergy was detected in these 12 patients as a result of their DPTs. Skin tests with penicillin DAP diagnostic sets could not be performed for all those with suspected beta-lactam allergies because DAP diagnostic sets were not always available at our hospital.

It was demonstrated that four patients exhibited grade 1 DHRs to DPTs when administered amoxicillin clavulanate, and a single patient demonstrated a grade 1 DHR to cefepime. Grade 1 DHRs were observed in three patients in response to quinolone group antibiotics, two patients in response to clindamycin, and in one patient in response to metronidazole during DPTs. In addition, grade 1 DHRs were seen to occur during DPTs with medications including methylprednisolone (*n* = 1), dexamethasone (*n* = 1), and pheniramine (*n* = 1), which are conventionally employed in the treatment of allergies. Detailed information on these DPTs is given in Figure 4.

## 4. Discussion

Suspected drug allergy is one of the most common problems encountered in practice in allergy and immunology clinics. Some patients think that they are allergic to a single drug, others to multiple drugs. The gold standard method for confirming drug allergy is to perform diagnostic tests with the suspected drugs. However, in many clinics, tests with alternative drugs are preferred due to inadequate facilities for conducting DPTs or patients’ refusal to accept diagnostic tests. Nevertheless, if diagnostic testing with the suspected drugs is performed in appropriately selected patients, the fear of drug allergy in that patient is eliminated. Our study is one of the largest studies on patients with suspected allergy in our country, and it includes the results of diagnostic drug tests performed in an adult clinic of a tertiary-care city hospital. A diagnosis of drug allergy was ruled out in 137 (73%) patients after DPTs. There are many conditions that cause drug allergy to be excluded in the majority of patients presenting with a suspected drug allergy. Most frequently, the patients consider the side effects of medication to be a drug allergy. Therefore, they may mislead physicians about their drug allergy. In addition, some physicians may have difficulty distinguishing between type A and type B drug reactions. Another reason may be that some skin rashes that are due to viral infections, especially those that our patients had in childhood, are mistakenly interpreted as drug allergies, and this may be a false label givent to the patient.

In our study, a DHR to a single suspected drug was observed in 78 of 188 patients. In contrast, the remaining 110 patients exhibited DHR histories involving multiple drugs. The rate of confirmation of drug allergy in diagnostic tests was higher in those who described a history of multiple drug allergy. Diagnostic tests are positive if there is a history of multiple drug allergies due to cross-reactions between beta-lactam antibiotics and cross-reactions between NSAIDs, especially those with a COX-1-mediated mechanism [13,16]. As in many studies, the groups of drugs most frequently suspected in our study were beta-lactam antibiotics, with NSAIDs being the second most frequently suspected [5,7,18,19,20]. The diagnosis of drug allergy was confirmed in only 8% of those with a suspected beta-lactam antibiotics allergy in this study. Despite the commonly held belief that beta-lactam allergies are the most common cause of DHRs, this study revealed that the actual sensitivity rate is, in fact, low. When the current guidelines and studies on beta-lactam antibiotics are evaluated, it is thought that performing diagnostic evaluations in patients with a low risk for anaphylaxis will both prevent unnecessary drug restrictions and reduce treatment costs because the use of more expensive alternative drugs will decrease [18,19,20,21].

Among the antibiotics tested, quinolones were the drugs most commonly suspected of causing allergy after beta-lactams in our study. The increasing frequency of use of quinolones over the years has affected this result. The usefulness of skin tests in diagnosing DHRs to quinolones is controversial, with their sensitivity and specificity varying between studies [22,23,24]. In our study, the diagnosis of allergy was confirmed in four out of five patients with a suspicion of quinolone allergy. We believe that detailed histories should be taken in patients with suspected quinolone allergy, the suspicion should be taken seriously, and a diagnostic evaluation should be performed.

In this study, NSAIDs were among the most common drugs to which patients expressed a suspicion of drug allergy. The diagnosis of drug allergy was confirmed through diagnostic testing in approximately one-quarter of the patients. Diagnostic drug test positivity was not related to the mechanisms of action (COX-1- vs. IgE-mediated) of the NSAID-induced DHRs. In a study conducted in children, a total of 238 DPTs were performed for suspected drugs, and 34 were positive [25]. In another study, NSAID DPTs were found to be positive in only one out of four patients [7]. Contrary to patients’ suspicions, the frequency of true drug allergy is low in NSAIDs, as it is in antibiotics. After an accurate and detailed assessment, the label of drug allergy can be removed from patients through diagnostic tests.

In the process of planning diagnostic tests for suspected drug allergies, consideration must be given to the characteristics of patients’ initial reactions. This is of particular importance in cases where patients have previously experienced intraoperative anaphylaxis. In such instances, the identification of the responsible drugs by means of diagnostic skin tests is imperative for the planning of future operations. Previous studies revealed that the most prevalent drugs causing intraoperative allergies were neuromuscular blocking agents, with rocuronium allergy being the most common among these agents [26,27]. In our study, rocuronium and then midazolam were found to be responsible for most DHRs to intraoperative anesthesia.

In emergency departments, steroids and first-generation antihistamines are two of the most commonly prescribed drugs for the treatment of allergic diseases [28]. Although rare, DHRs to these drugs may also occur, and many physicians may not consider these drugs to be responsible for those reactions. Cases of anaphylaxis and drug allergy to antihistamines or steroids have been reported in the literature [29,30,31]. In our study, two patients were confirmed to have pheniramine allergy through IDTs and one patient had this confirmed via a DPT. It is imperative to consider DHRs related to steroids and antihistamines and to clarify drug allergies with diagnostic tests when necessary. In particular, those who are allergic to these drugs, which are frequently used in emergency departments in our country, should be recorded in the National Health Information System.

In our study, suspected drug allergy was seen mostly in women and in young-to-middle-aged patients. As in many studies conducted on allergic diseases, dominance of the female sex was observed [5,18,19,32,33,34]. In studies on occupations and drug allergy, the risk was found to be high in health workers and those working in drug production [35,36]. In our study, the drug allergy of three healthcare workers was confirmed through diagnostic drug tests. In a study conducted in Portugal, 7.7% of university students reported drug allergies [37]. In our study, although about one-third of patients were housewives, and diagnostic drug tests were mostly negative in students. We interpreted the reason for the low actual sensitization of students who reported a suspicion of drug allergy as being due to the fact that students could access misinformation about drug allergies on the internet and exaggerate their symptoms to physicians.

Our study is one of the first to provide diagnostic evaluation results in adult patients with suspected drug allergies in a tertiary-care city hospital. Nevertheless, this study is not without its limitations. The first limitation is that the number of patients who underwent diagnostic drug testing was lower than the number of patients who underwent drug testing for alternative drugs. This is because our allergy clinic was recently established (2019) and part of the study coincided with the COVID-19 pandemic period. The second limitation of our study is the difficulties experienced in the availability of standardized penicillin skin test kits, which have a very important place in the diagnostic evaluation of beta-lactam antibiotics, in our hospital. Because of this, we think that allergy physicians had to perform tests with alternative drugs instead of diagnostic tests. The third limitation is that some patients refused to undergo diagnostic drug testing, even though they were at low risk from the diagnostic evaluation. Although the number of patients included in this study was higher than the number of patients that needed to be included according to our power analysis, future prospective studies in a larger patient group are needed to draw stronger conclusions.

## 5. Conclusions

In conclusion, contrary to the prevailing notion that drugs (especially beta-lactams) are the predominant cause of allergic reactions, this study has demonstrated that the actual prevalence of drug allergies is, in fact, low. It is vital for the future treatment of patients to confirm or rule out the diagnosis of drug allergy by taking a detailed anamnesis, correctly determining the risks of diagnostic drug testing, and selecting appropriate diagnostic methods.

## Figures and Tables

**Figure 1 medicina-61-00386-f001:**
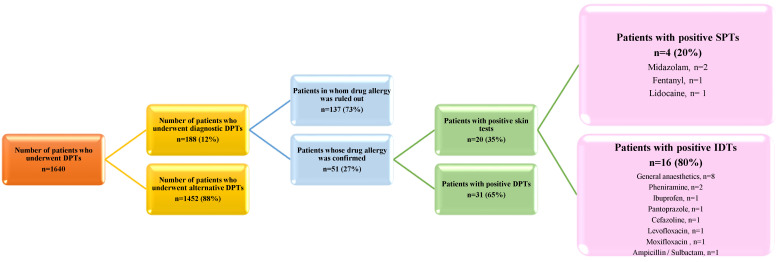
The study design and results of drug skin tests. DPT, drug provocation test; IDT, intradermal test; SPT, skin prick test.

**Figure 2 medicina-61-00386-f002:**
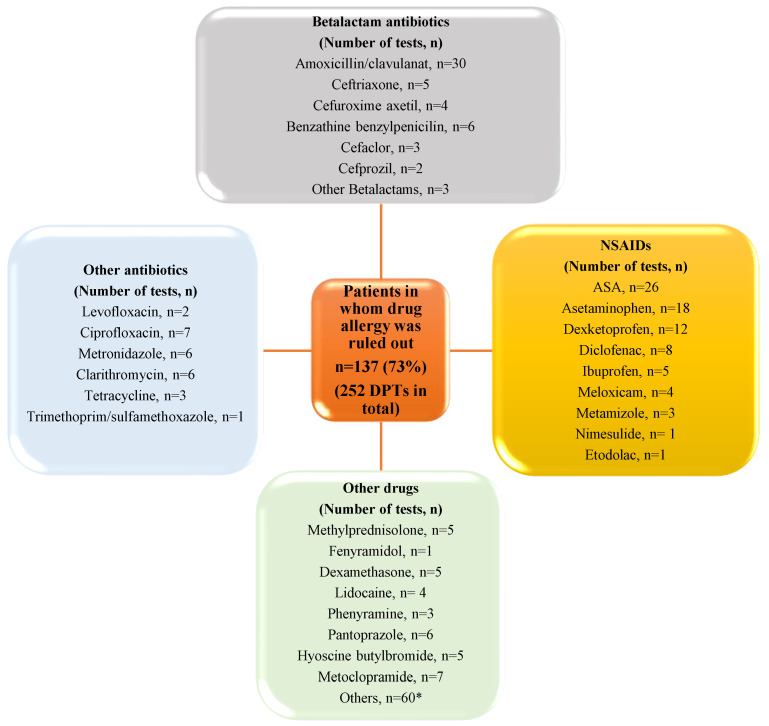
Results of negative drug provocation tests. ASA, Acetyl salicylic acid; DPT, drug provocation test. * Vitamins, iron drugs, thyroid drugs and etc.

**Figure 3 medicina-61-00386-f003:**
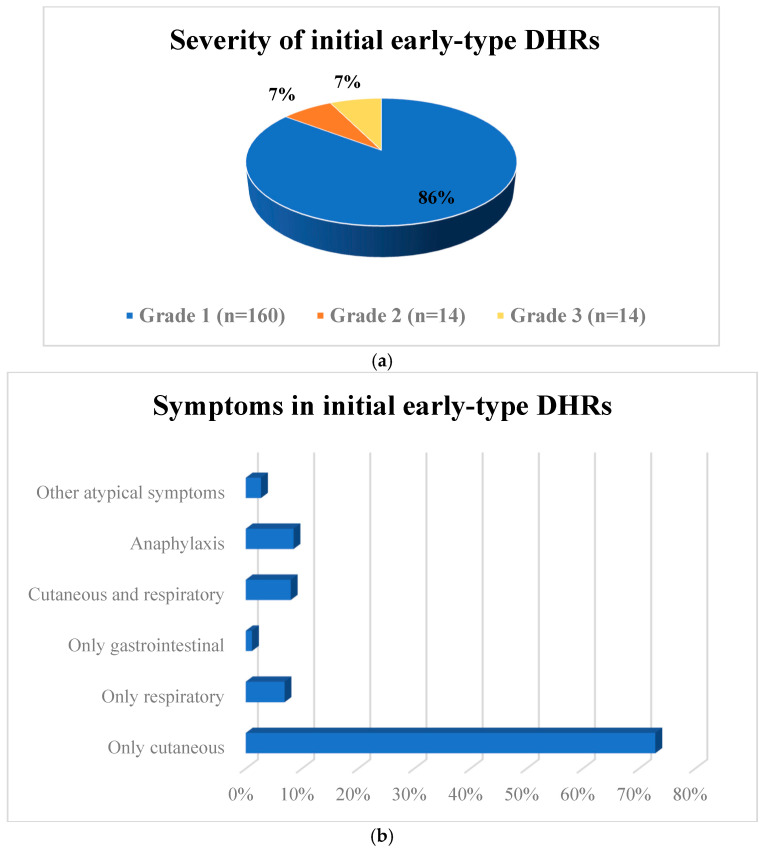
(**a**) Severity of initial early-type DHRs. (**b**) Symptoms in initial early-type DHRs. DHRs, drug hypersensitivity reactions.

**Figure 4 medicina-61-00386-f004:**
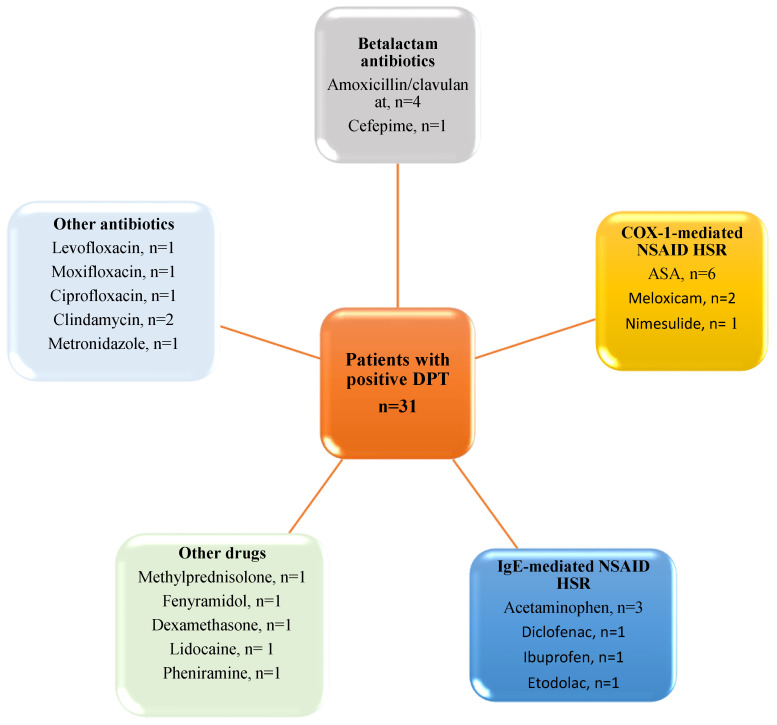
Results of positive drug provocation tests. ASA, Acetyl salicylic acid; COX, Cyclooxygenase; DHR, drug hypersensitivity reaction; DPT, drug provocation test; NSAIDs, non-steroidal anti-inflammatory drugs.

**Table 1 medicina-61-00386-t001:** Comparison of diagnostic drug test results in terms of demographic and clinical characteristics.

Parameters	Diagnostic Drug Test Results		*p*-Value
	Negative *(n* = 137)	Positive (*n* = 51)	
**Age (years) [median (min–max)]**	37 (18–70)	43 (21–65)	0.147
**Sex [*n* (%)]**			0.327
Female	108 (74)	38 (26)	
Male	29 (69)	13 (31)	
**Occupation [*n* (%)]**			0.032 *
Housewife	48 (72)	19 (28)	
Civil servant	32 (65)	17 (35)	
Employee	12 (80)	3 (20)	
Retired	18 (64)	10 (36)	
Student	27 (93)	2 (7)	
**Education [*n* (%)]**			0.616
Primary school	15 (75)	5 (25)	
Secondary school	21 (70)	9 (30)	
High school	37 (67)	18 (33)	
University	64 (77)	19 (23)	
**Comorbidities [*n* (%)]**			0.442
Cardiovascular	8 (89)	1 (11)	
Endocrine	7 (78)	2 (22)	
Rheumatologic	8 (73)	3 (27)	
Gastroenterologic	11 (58)	8 (42)	
Malignancy	1 (50)	1 (50)	
Allergic and immunologic	40 (75)	13 (25)	
**History of allergic diseases [*n* (%)]**			0.491
Asthma	2 (50)	2 (50)	
Allergic rhinitis	10 (77)	3 (23)	
Urticaria	26 (84)	5 (16)	
Nasal polyps	2 (40)	3 (60)	
**Atopy [*n* (%)]**	34 (76)	11 (24)	0.704
**Drug allergy suspicion [*n* (%)]**			0.046 *
Single drug	63 (81)	15 (19)	
Multiple drugs	74 (67)	36 (33)	
**Number of suspected drugs [median (min–max)]**	1 (1–5)	2 (2–5)	0.010 *
**Type of suspected drug [*n* (%)]**			0.140
Antibiotic	46 (75)	15 (25)	
NSAID	46 (77)	14 (23)	
Antibiotic and NSAID	16 (84)	3 (16)	
PPI	2 (67)	1 (33)	
Antibiotic and PPI	4 (100)	0	
Local anesthetic drug	4 (67)	2 (33)	
Other drugs (e.g., general anesthetics)	19 (54)	16 (46)	
**Type of suspected antibiotic [*n* (%)]**			0.002 *
Beta-lactam	43 (92)	4 (8)	
Quinolone	1 (20)	4 (80)	
Macrolide	1 (50)	1 (50)	
Others	3 (75)	1 (25)	
Multiple antibiotics	16 (73)	6 (27)	
**Type of suspected beta lactam [*n* (%)]**			0.615
Penicillin G or V	6 (100)	0	
Amoxicillin/clavulanate	30 (86)	5 (14)	
Cephalosporin	14 (88)	2 (12)	
**Possible mechanism of suspected NSAID [*n* (%)]**			0.414
IgE-mediated	25 (83)	5 (17)	
COX-1	37 (74)	13 (26)	

* *p* < 0.05 was considered significant. Cyclooxygenase, COX; Immunoglobulin E, IgE; minimum, min; maximum, max; non-steroidal anti-inflammatory drugs, NSAIDs; Proton pump inhibitors, PPIs.

**Table 2 medicina-61-00386-t002:** Comparison of patient characteristics in terms of their initial drug hypersensitivity reactions.

Parameters	Result of Diagnostic Drug Test		*p*-Value
	Negative (*n* = 137)	Positive (*n* = 51)	
**Severity of initial early-type DHRs [*n* (%)]**			0.001 *
Grade 1	129 (81)	31 (19)	
Grade 2	7 (50)	7 (50)	
Grade 3	1 (7)	13 (93)	
**At what dose of the drug [median (min–max)]**	1 (1–10)	1 (1–10)	0.052
**At what minute after the drug was administered [median (min–max)]**	60 (2–600)	15 (2–300)	0.001 *
**Itching [*n* (%)]**			0.042 *
Yes	79 (68)	38 (32)	
No	58 (82)	13 (18)	
**Redness [*n* (%)]**			0.001 *
Yes	29 (54)	25 (46)	
No	108 (81)	26 (19)	
**Urticaria [*n* (%)]**			0.001 *
Yes	45 (56)	35 (44)	
No	92 (85)	16 (15)	
**Angioedema [*n* (%)]**			0.031 *
Yes	35 (61)	22 (39)	
No	102 (78)	29 (22)	
**Dyspnea-bronchospasm [*n* (%)]**			0.001 *
Yes	11 (37)	19 (63)	
No	126 (80)	32 (20)	
**Cyanosis-desaturation [*n* (%)]**			0.001 *
Yes	0	12 (100)	
No	137 (78)	39 (22)	
**Syncope [*n* (%)]**			0.001 *
Yes	1 (11)	8 (89)	
No	136 (76)	43 (24)	
**Arrhythmia-tachycardia [*n* (%)]**			0.001 *
Yes	3 (18)	14 (82)	
No	134 (78)	37 (22)	
**Hypotension [*n* (%)]**			0.001 *
Yes	0	5 (100)	
No	137 (75)	46 (25)	
**Feeling of sickness [*n* (%)]**			0.030 *
Yes	10 (50)	10 (50)	
No	127 (76)	41 (24)	
**Diarrhea–vomiting [*n* (%)]**			0.082
Yes	9 (53)	8 (47)	
No	128 (75)	43 (25)	
**Treatment of initial DHR [*n* (%)]**			0.001 *
Only antihistamine	23 (82)	5 (18)	
Antihistamine and steroid	66 (70)	29 (30)	
Antihistamine, steroid, and adrenaline	0	12 (100)	
**Symptoms with placebo during DPT [*n* (%)]**			0.007 *
Yes	16 (100)	0	
No	121 (70)	51 (30)	

* *p* < 0.05 was considered significant. DHR, drug hypersensitivity reaction; DPT, drug provocation test; minimum, min; maximum, max.

## Data Availability

The authors declare that they have followed the protocols of their work center for the publication of patient data in this study. All data generated or analyzed during this study are included in this article. The data that support the findings of this study are available on request from the corresponding author. The data are not publicly available due to privacy or ethical restrictions.

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
