# Peer review of "Is a Drug Allergy in a Patient’s History Real? Our Experience with Diagnostic Drug Provocation Tests"

_medicina, 2025, doi:10.3390/medicina61030386_

Round 1
Reviewer 1 Report
Comments and Suggestions for Authors
The manuscript "Is Drug Allergy in Patient’s History Real? Our Experience with Diagnostic Drug Provocation Tests" is a retrospective study assessing demographic characteristics, responsible medications, types of drug hypersensitivity reactions (DHRs), and results of diagnostic drug provocation tests (DPs) in patients with suspected drug allergy. The study included 188 adults who underwent diagnostic testing for drug allergy. The authors aimed to evaluate the reliability of indications of allergy in the patients’ medical history.
The authors need to correct the following comments or provide justified responses:
1) 188 patients is a relatively small sample size for a retrospective study, especially considering the variety of drugs and types of reactions. The statistical power of the study needs to be clarified.
2) The reviewed text poorly reflects the frequencies of positive and negative DPs for different groups of drugs. It does not indicate how many patients had a reaction to each specific drug. This information is necessary to assess the significance of the obtained results.
3) The main question of the study is the agreement between the anamnesis and the results of the DPs. The reviewed text does not analyze these data. It is necessary to show the percentage of cases where the anamnesis was not confirmed by the DPs, and discuss possible reasons for such discrepancies.
4) The discussion should thoroughly consider the limitations of the study, including the retrospective design, sample size, and the possibility of systematic errors in the analysis of medical records.
Author Response
Reviewer #1
The manuscript "Is Drug Allergy in Patient’s History Real? Our Experience with Diagnostic Drug Provocation Tests" is a retrospective study assessing demographic characteristics, responsible medications, types of drug hypersensitivity reactions (DHRs), and results of diagnostic drug provocation tests (DPs) in patients with suspected drug allergy. The study included 188 adults who underwent diagnostic testing for drug allergy. The authors aimed to evaluate the reliability of indications of allergy in the patients’ medical history. The authors need to correct the following comments or provide justified responses:
Comment 1.
188 patients is a relatively small sample size for a retrospective study, especially considering the variety of drugs and types of reactions. The statistical power of the study needs to be clarified.
Response 1. Thank you very much for allowing us to improve our manuscript and evaluating our manuscript for publication in Medicina.
In this study, power analysis was conducted utilising G Power 3.1, and sample analysis was performed. The following parameters were configured for the power analysis: effect size = 0.5, alpha error probability = 0.05, power (1-B error probability) = 0.8, and allocation ratio N2/N1= 3. The total sample size was calculated to be 170 (43 in the first group and 127 in the second group).
We added this as “In this study, power analysis was conducted utilising G Power 3.1, and sample analysis was performed. The following parameters were configured for the power analysis: effect size = 0.5, alpha error probability = 0.05, power (1-β error probability) = 0.8, and allocation ratio N2/N1= 3. The total sample size was calculated to be 170’’ in lines 166-169 in the revised clean copy of the manuscript.
Comment 2.
The reviewed text poorly reflects the frequencies of positive and negative DPs for different groups of drugs. It does not indicate how many patients had a reaction to each specific drug. This information is necessary to assess the significance of the obtained results.
Response 2.
Thank you very much for your insightful comments and suggestions.
A total of 1640 patients underwent alternative or diagnostic DPTs in allergy and clinical immunology clinics between February 2019 and December 2024. A total of 188 patients underwent diagnostic drug allergy testing to confirm or exclude their drug allergy. After the diagnostic evaluation, the diagnosis of drug allergy was confirmed in 51 (27%) patients. A diagnosis of drug allergy was ruled out in 137 (73%) patients after DPTs. DPTs were performed with a total of 252 drugs in 137 patients in whom drug allergy was ruled out. Among these patients, amoxicillin clavunate was the most frequently performed drug, and a total of 30 DPTs were performed. The second most frequently administered DPTs drug was ASA (n=26). The detailed results are shown in detail in Figure 2.
Figure 2. Results of negative drug provocation tests. ASA, Acetyl salicylic acid; DPT, Drug provocation test
We added this as “DPTs were performed with a total of 252 drugs in 137 patients in whom drug allergy was ruled out. Among these patients, amoxicillin clavunate was the most frequently performed drug, and a total of 30 DPTs were performed. The second most frequently drug was ASA (n=26). The detailed results are shown in detail in Figure 2.’ in lines 185-188 in the revised clean copy of the manuscript.
Comment 3.
The main question of the study is the agreement between the anamnesis and the results of the DPs. The reviewed text does not analyze these data. It is necessary to show the percentage of cases where the anamnesis was not confirmed by the DPs, and discuss possible reasons for such discrepancies.
Response 3.
Thank you very much for your insightful comments and suggestions.
A total of 1640 patients underwent alternative or diagnostic DPTs in allergy and clinical immunology clinics between February 2019 and December 2024. A total of 188 patients underwent diagnostic drug allergy testing to confirm or exclude their drug allergy. After the diagnostic evaluation, the diagnosis of drug allergy was confirmed in 51 (27%) patients. A diagnosis of drug allergy was ruled out in 137 (73%) patients after DPTs. There are many conditions that cause drug allergy to be excluded in the majority of patients presenting with suspected drug allergy. Most importantly, the patients consider the side effects of medication as drug allergy. Therefore, they may mislead physicians about drug allergy. In addition, some physicians may have difficulty distinguishing between type A and type B drug reactions. Another reason may be that some skin rashes due to viral infections, especially those that our patients had in childhood, are mistakenly interpreted as drug allergies, and this may be a false label left on the patient.
We added a part as follows ‘A diagnosis of drug allergy was ruled out in 137 (73%) patients after DPTs. There are many conditions that cause drug allergy to be excluded in the majority of patients presenting with suspected drug allergy. Most importantly, the patients consider the side effects of medication as drug allergy. Therefore, they may mislead physicians about drug allergy. In addition, some physicians may have difficulty distinguishing between type A and type B drug reactions. Another reason may be that some skin rashes due to viral infections, especially those that our patients had in childhood, are mistakenly interpreted as drug allergies, and this may be a false label left on the patient.’ in lines 359-366 in the revised clean copy of the manuscript.
Comment 4. The discussion should thoroughly consider the limitations of the study, including the retrospective design, sample size, and the possibility of systematic errors in the analysis of medical records.
Response 4. Thank you very much for your constructive comments and contributions.
Our study is one of the first to provide diagnostic evaluation results in adult patients with suspected drug allergies in a tertiary care city hospital. Nevertheless, this study is not without its limitations. The first limitation is that the number of patients who underwent diagnostic drug testing was lower than the number of patients who underwent drug testing for alternative drugs. This is because our allergy clinic was recently established (2019) and part of the study coincided with the COVID-19 pandemic period. The second limitation of our study is the difficulties experienced in the availability of the standardized penicillin skin test kits, which have a very important place in the diagnostic evaluation of beta-lactam antibiotics, in our hospital. Therefore, it was thought that allergy physicians preferred to perform tests with alternative drugs instead of diagnostic tests. The third limitation is that some patients refused to undergo diagnostic drug testing, even though they were at low risk for diagnostic evaluation. In addition, the retrospective nature of the study is another limitation of the study. Although the number of patients included in this study was higher than the number of patients who should have been included according to the power analysis, future prospective studies in a larger patient group are needed to make stronger interpretations. Since the medical records of the patients in our clinic are stored with both the file system and the hospital information management system, the possibility of systematic errors is very low.
In the revised clean copy of the article line 441-444 we have added a section as follows.
‘Although the number of patients included in this study was higher than the number of patients who should have been included according to the power analysis, future prospective studies in a larger patient group are needed to make stronger interpretations.‘

Reviewer 2 Report
Comments and Suggestions for Authors
The Authors report their retrospective experience on a population of 188 adults with suspected drug allergies in a tertiary care Hospital, who underwent diagnostic drug testing. Diagnosis of drug allergy was confirmed in 27% of cases, while in 73% of cases drug allergy was excluded. A drug hypersensitivity reaction (DHR) was present in 78 patients while the other 110 had the diagnosis of drug allergy excluded after a drug provocation test. In 78/188 patients, there was a DHR to a single suspected drug while 110/188 had DHR with multiple drugs.
The study design is well presented and the statistical analysis performed appropriately chosen.
The Figures are highly explicative of the presented data.
The study results are of interest as they underline that performing diagnostic evaluations in patients with a low risk of anaphylaxis can prevent unnecessary drug restrictions or the use of possible more expensive alternative drugs.
The paper includes also data on patients with a history of intraoperative anaphylaxis and DHRs to first generation antihistamines and steroids, giving hints on how to deal with these clinical cases.
Data from the population studied present also gender and age related characteristics related to DHRs, as drug allergy was seen mostly in women and in young to middle aged patients.
The Authors at the end of the paper underline some limitations of the current study even if these considerations potentiate the study without reducing its rigor.
Overall the paper is well written and adds real-world data on DHR from a monocentric tertiary care setting.
Author Response
Thank you very much for allowing us to improve our manuscript and evaluating our manuscript for publication in Medicina.
